# Genomics of Viral Hepatitis-Associated Liver Tumors

**DOI:** 10.3390/jcm10091827

**Published:** 2021-04-22

**Authors:** Camille Péneau, Jessica Zucman-Rossi, Jean-Charles Nault

**Affiliations:** 1Centre de Recherche des Cordeliers, Sorbonne Université, Inserm, Université de Paris, F-75006 Paris, France; peneaucam@gmail.com (C.P.); jessica.zucman-rossi@inserm.fr (J.Z.-R.); 2Functional Genomics of Solid Tumors Laboratory, Équipe Labellisée Ligue Nationale Contre le Cancer, Labex OncoImmunology, F-75006 Paris, France; 3Hôpital Européen Georges Pompidou, APHP, F-75015 Paris, France; 4Service d’hépatologie, Hôpital Avicenne, Hôpitaux Universitaires Paris-Seine-Saint-Denis, Assistance-Publique Hôpitaux de Paris, F-93000 Bobigny, France; 5Unité de Formation et de Recherche Santé Médecine et Biologie Humaine, Université Paris Nord, F-93000 Bobigny, France

**Keywords:** hepatitis B, adeno associated virus type 2, genetic alterations, hepatitis delta

## Abstract

Virus-related liver carcinogenesis is one of the main contributors of cancer-related death worldwide mainly due to the impact of chronic hepatitis B and C infections. Three mechanisms have been proposed to explain the oncogenic properties of hepatitis B virus (HBV) infection: induction of chronic inflammation and cirrhosis, expression of HBV oncogenic proteins, and insertional mutagenesis into the genome of infected hepatocytes. Hepatitis B insertional mutagenesis modifies the function of cancer driver genes and could promote chromosomal instability. In contrast, hepatitis C virus promotes hepatocellular carcinoma (HCC) occurrence mainly through cirrhosis development whereas the direct oncogenic role of the virus in human remains debated. Finally, adeno associated virus type 2 (AAV2), a defective DNA virus, has been associated with occurrence of HCC harboring insertional mutagenesis of the virus. Since these tumors developed in a non-cirrhotic context and in the absence of a known etiological factor, AAV2 appears to be the direct cause of tumor development in these patients via a mechanism of insertional mutagenesis altering similar oncogenes and tumor suppressor genes targeted by HBV. A better understanding of virus-related oncogenesis will be helpful to develop new preventive strategies and therapies directed against specific alterations observed in virus-related HCC.

## 1. Introduction

Eliminating viral hepatitis has become a priority goal for the World Health Organization, which estimates that today, 325 million people are globally living with a viral hepatitis worldwide, with a vast majority being undiagnosed and/or untreated [1]. The viral cause of hepatitis was described for the first time in 1944 [2]. From 1965 to 1989, five hepatotropic viruses were identified—Hepatitis Viruses A, B, C, D, and E (Table 1) [3]. Several other risk factors such alcohol consumption or metabolic liver diseases can induce liver inflammation but viral infections remain so far the leading cause of chronic liver disease worldwide [4]. Hepatitis B, C, and D viruses can induce chronic infection; among people infected by Hepatitis B Virus (HBV), more than 90% of neonates develop chronic hepatitis but less than 10% in adulthood [5]. Due to the frequent progression of chronic hepatitis to cirrhosis and liver tumor development, chronic hepatitis B and C infections account for 1.3 million deaths each year [1].

Liver tumor is currently the fourth cause of death by cancer in the world with 782,000 deaths per year [6]. Hepatocellular carcinoma (HCC) represents 90% of all primary liver tumors and is related to HBV or Hepatitis C Virus (HCV) chronic infections in 85% of the cases [7]. In HBV- and HCV-infected patients, liver disease progression is explained by the constant and unsuccessful attempts of the immune system to clear the virus, leading to chronic liver damage [8]. Regardless of the cause (viral infection, chronic alcohol consumption, metabolic syndrome), chronic inflammation leading to liver cirrhosis plays a crucial role in carcinogenesis as up to 85% of HCC occurs on a background of cirrhosis [9]. Early oncogenic steps include WNT-β-catenin pathway activation, re-expression of fetal genes, deregulation of protein-folding machinery and the response to oxidative stress, replicative senescence bypassed by TERT promoter mutation [9]. However, apart from this common indirect mechanism, some studies showed that hepatitis viruses may have direct oncogenic properties, and especially HBV as 10 to 30% of HBV-associated HCC develop in a non-cirrhotic liver [10,11].

HCC are highly heterogenous tumors, and the knowledge of their genetic diversity has significantly increased in the recent years with advances in molecular classification and dissection of genetic drivers [11,12]. The main somatic genetic alterations observed in HCC were related to telomere maintenance (*TERT* promoter 40–60%, *TERT* amplification 5%), cell cycle control (*TP53* 15–40%, *RB1* 3–8%, *CDKN2A* 2–12%), Wnt-β catenin pathway (*CTNNB1* 10–35%, *AXIN1* 5–15%), epigenetic regulation (*ARID1A* 5–15%, *ARID2* 3–10%), oxidative stress pathway (*NFE2L2* 3–6%, *KEAP1* 2–8%) and AKT/mTOR and Ras/Raf MAP Kinase pathways (*RPS6KA3* 2–9%, *PIK3CA* 1%, *TSC1* 2%, *TCS2* 3%, *FGF19* 5%) with 2 to 6 driver genes altered per tumor among a total of 40 to 60 coding somatic mutations [13,14,15,16,17]. These genomic studies have highlighted specificities related to risk factors and helped to better understand the role played by viruses in hepatocarcinogenesis.

This review aims to give an overview about the genetics of viral hepatitis-associated liver tumors (Table 2) and the direct and indirect consequences of viral infections on HCC development.

## 2. Genomics of Hepatitis B-Related HCC

More than 50% of HCC worldwide are associated with HBV infection and this percentage rises to 85% in some areas where the virus is endemic (China, Sub-Saharan Africa) [18]. Moreover, these numbers are probably underestimated because some HCC may develop as part of occult HBV infections that are usually not identified [19]. The risk of developing HCC is 15 to 20 times higher in people with chronic B infection than in the uninfected population [18]. This risk increases with the progression of liver disease: among cirrhotic patients, the incidence of developing HCC over 5 years is 15% in highly endemic regions such as Asia and Africa and 10% in the rest of the world [10]. However, regardless of the severity of the underlying liver disease, the risk of HCC in HBV-infected individuals is higher in endemic areas than in Western countries, possibly due to early acquisition of the virus at birth, longer duration of infection, or exposure to environmental genotoxic toxins such as aflatoxin B1 and aristolochic acid.

### 2.1. HBV Structure and Cell Cycle

Hepatitis B Virus (HBV) is a DNA virus of 3.2 kb. In an infectious viral particle, the HBV DNA is circular and partially double-stranded, in a form known as relaxed circular DNA (rcDNA). The HBV structure relies on an envelope composed of lipids and three types of surface proteins (HBsAg), containing a nucleocapsid composed of dimers (HBcAg), inside which the viral DNA genome is covalently linked with the viral polymerase [20].

Once entered in the cell via endocytosis, the surface proteins are cleaved, allowing the release of the nucleocapsid, which is transported to the nucleus [21]. The HBV rcDNA is then converted into a persistent covalently closed circular DNA (cccDNA) form compacted with cellular proteins (histones and non-histones) into a mini-chromosome [22]. Viral messenger RNAs (mRNA) are transcribed from HBV cccDNA and transported to the cytoplasm to be translated by the host cell in the different viral proteins, including HBx, a protein essential for cccDNA transcription [23]. HBV pregenomic RNA (pgRNA) is packaged by the capsid proteins and retrotranscribed by the viral polymerase into rcDNA (in 90% of cases) or into a double-strand linear DNA (dslDNA) form (in 10% of cases) [24]. Part of the nucleocapsid containing rcDNA or dslDNA returns to the host cell nucleus to be recycled and thus allows the maintenance of a pool of cccDNA in the nucleus. The other part is enveloped with the surface proteins and released outside the cell. Inside the nucleus, dslDNA may be integrated in the host genome. This illegitimate integration process is independent of the virus replication cycle and does not depend on the expression of viral proteins [25].

### 2.2. Genomic Landscape of HBV-Related HCC

Genomic alterations as well as transcriptomic dysregulation observed in HCC are strongly associated with risk factors. A recent study compared the mutation and transcriptomic profiles (reviewed in [26]) of HBV-related HCC with HCC related to other etiologies. HBV-related HCC were enriched in proliferative subclasses (G1-G3 transcriptomic subgroups) and specially in HCC with stem cell features (G1 transcriptomic subgroup). Moreover, the cell cycle pathway was more frequently altered in HBV-related HCC, mainly via inactivating mutations of *TP53* (41% in HBV versus 16% in other etiologies) and *IRF2*, these two mutations being mutually exclusive [27]. In contrast, mutations of *NFE2L2* or *CTNNB1* genes were less frequent in HBV-related HCC [12]. In these HBV-related tumors, the mutational spectrum of *TP53* has the particularity to have a high frequency of mutations on codon 249 (R249S) caused by exposure to Aflatoxin B1, as discussed below. Furthermore, *TP53* mutations are associated with lower survival among patients with HBV-related HCC but not among those with HCC related to another etiology, suggesting a prognostic value of these mutations only in this specific group of particularly aggressive HCC [27].

Moreover, one of the genomic feature of HBV-related HCC is the presence of frequent viral integrations in genes involved in carcinogenesis such as *TERT*, *MLL4*, and *CCNE1* [11]. Finally, a higher rate of chromosomal instability is observed in HBV-related HCC [28]. This can probably be explained by the ability of HBV to induce genomic instability via viral integrations and HBx activity as well as by the frequency of mutations observed in *TP53* and *IRF2*, two genes involved in the maintenance of chromosome stability (Figure 1) [11].

### 2.3. HBV Oncogenic Proteins

The HBx protein contains 154 amino acids and is encoded by the smallest open reading frame (ORF) of the HBV viral genome. This protein has been studied extensively and the results obtained vary considerably depending on the cell models and experimental conditions used. As experiments have been performed with a strong overexpression of the HBx protein or in the absence of other viral proteins, they do not necessarily reflect the infectious context [29]. The functions of this protein are still controversial but can be broadly divided into two groups: the role of HBx in viral replication through degradation of the host factors and transcriptional modulation and its involvement in hepato-carcinogenesis.

HBx could foster tumor development through the following mechanisms (reviewed in [11,30,31]): (1) transcription dysregulation due to its interaction with transcription factors, chromatin-modifying enzymes, or components of the cellular transcriptional machinery, (2) regulation of senescence through *TERT* overexpression, inhibition of p53 or Rb proteins, and p53 sequestration in the cytoplasm [31], (3) regulation of apoptosis through both pro- and anti-apoptotic properties, depending on the level of HBx expression and the cellular context, (4) impairment of DNA repair through interaction with DDB1 or p53 proteins, (5) dysregulation of the cell cycle and the mitotic process through activation of MAP-Kinase, JNK, or Src kinase pathways.

In contrast, a recent study reported a higher frequency of inactivating mutations of the HBx gene in HCC compared to non-tumor tissues, suggesting a role of tumor suppressor gene of HBx and the existence of a selection pressure during carcinogenesis that could promote proliferation of cells in which HBx is inactive [27].

### 2.4. HBV Integrations

The identification of an HBV sequence integrated into the DNA of a human HCC was first described in 1980 [32] and it is now admitted that HBV integration events are observed in almost all tumors (80–90%) [33]. However, since integrations occur within a few days after infection [34], decades may elapse between viral insertion and HCC development. So far, three mechanisms have been proposed to explain the oncogenic consequences of integrations: insertional mutagenesis nearby a cancer-related gene, expression of abnormal viral proteins from integrated HBV DNA, and induction of chromosomal instability [24,35].

First, viral integrations may induce changes in gene expression via activation of transcription through integration in promoters, exons or introns, and, to a lesser extent, in intergenic regions [36,37,38]. Three genes have been identified as recurrent integration sites (40% of HBV-related HCC): *TERT*, *MLL4*, and *CCNE1* [39,40]. *TERT* is the gene the most frequently altered by HBV integrations that are mainly located in the promoter and mutually exclusive of other *TERT* alterations such as promoter mutations [14,41,42]. Moreover, the closer the integration site is to the transcription initiation site, the higher is *TERT* expression, suggesting that the viral sequence acts as an enhancer [38]. Two recent studies using cellular models have shown that the “Enhancer I” region can induce *TERT* activation via the recruitment of transcription factors [40,42]. *MLL4* integrations are located between exons 2 and 6 and induce the synthesis of fusion HBV-MLL4 transcripts. HBV integrations generally modify the open-reading frame, resulting in a loss of function [39,43,44,45]. Finally, viral insertions in *CCNE1* induce an overexpression of the gene and a deregulation of the cell cycle fostering replicative stress responsible of a specific signature of rearrangements in the tumor genome [46]. The same genomic pattern is observed in HCC with HBV integrations in *CCNA2* leading to a protein truncated at its N-terminal end, lacking of its regulatory domain. All HCC harboring an alteration in *CCNA2* or *CCNE1* (due to viral integration or structural rearrangements) belong to a homogeneous subgroup of large and aggressive tumors with a poor prognostic mostly developed in the absence of cirrhosis. A recent study showed that tumors with a high number of HBV integrations were associated with poor prognosis [47].

Moreover, the HBV sequence integrated in the hepatocyte genome could contain the full preS/S ORF region and its promoter and could be responsible of the expression of viral proteins. Several studies showed that HBsAg can derive from HBV integrated DNA [48,49] and remain functional as the formation of infectious HDV virions was observed [50,51]. Interestingly, HCC replicating the virus and expressing pgRNA were associated with a specific transcriptomic profile and have a good prognosis [49]. However, deletions of the preS region are frequently identified in HCC, removing the C-terminal part of preS1 domain or the N-terminal part of preS2 domain. In vitro, the expression of altered envelope proteins induces endoplasmic reticulum stress, oxidative stress, and DNA damage, leading to cell cycle progression and malignant transformation [52,53]. In addition, given the structure of dslDNA, the ORF X is truncated at its C-terminal end in integrated sequences, as observed in at least half of the HCC for which this viral gene was detectable [42]. This leads to the synthesis of a truncated or fusion transcript. The overexpression of a truncated HBx protein has been associated in in vitro and in vivo models with progenitor cell properties, malignant transformation, inhibition of apoptosis, and tumor invasion [42,54,55]. However, these phenotypes may be due to a HBx overexpression at a level much higher than its expression under the activity of its native promoter. Overall, as mentioned above, since the role played by the wild-type HBx protein in carcinogenesis is not well established, it is difficult to conclude on the oncogenic potential of the truncated HBx protein without alternative models of infection [56].

Finally, HBV integrations are enriched near fragile sites, repeated regions, CpG islands and telomeres in tumors but not in non-tumor tissues, suggesting that hepatocytes with alterations in these regions are selected during carcinogenesis and that such integrations promote HCC development by inducing genomic instability [57]. In addition, a significant proportion of the viral integration sites occurred in the vicinity of DNA copy number alterations [14,37,39]. We recently described frequent chromosome rearrangements at HBV integration sites leading to cancer-driver genes (*TERT*, *TP53*, *MYC*) alterations [47]. The number of integrations is also associated with the number of structural variants [41], and two hypotheses have been proposed to explain this association. Chromosomal instability induced by an external factor may promote the integration mechanism of the virus as HBV insertions occur preferentially at double-stranded break sites [41]. At the other side, viral integration can be a driver inducing genomic instability and structural variants, as discussed above for integrations in *CCNA2* or *CCNE1* genes [46].

## 3. Genomics of Hepatitis C-Related HCC

Hepatitis C Virus (HCV) is one of the main etiology of HCC in low incidence areas (Western Europe, North America, Japan, Middle-East) [7,18]. The risk of developing HCC in patients with chronic hepatitis C increases with the severity of fibrosis stage. Overall, the rate of HCC occurrence is 1–5% per year in cirrhotic patients [18] modulated by numerous factors related to host (older age, male sex, severity of liver disease, presence of comorbidities) or virus (HCV genotype, HCV viral load, coinfections with HBV or HIV, viral eradication) [10].

### 3.1. HCV Structure and Cell Cycle

HCV is a single-stranded, positive-sense linear RNA virus of 9.6 kb that includes a 5’ non-coding region, an open reading frame, and a 3’ non-coding region [58]. Viral particles are composed of a lipidic envelope with two viral envelope glycoproteins anchored (E1 and E2), containing an icosahedral capsid formed by Core proteins, inside which is the viral genome [59].

When the virion enters the cell, the fusion of the viral envelope to the endosome membrane and the dissociation of the viral core allow the release of the HCV genome in the cytoplasm [60]. The viral RNA is recognized as a mRNA and translated by the host cellular machinery to form a precursor polyprotein of about 3000 amino acids [61]. This polyprotein is then cleaved by cellular and viral enzymes into ten viral proteins: the structural proteins (Core protein, E1 and E2), the non-structural proteins (NS2, NS3, NS4A, NS4B, NS5A et NS5B), and the protein p7, which is the result of an incomplete cleavage from the E2 protein [62]. Replication complexes are then formed with non-structural proteins and viral RNA to catalyze the transcription of a negative-sense RNA intermediate from which the positive-sense RNA is finally generated. Capsid proteins assemble and recruit HCV genome, before being enveloped, and mature virions are released outside the cell via the secretory pathway [61].

### 3.2. Genomic Landscape of HCV-Related HCC

Compared to HBV-related HCC, HCV-related HCC are less aggressive, with chromosomal stability, more differentiated and they tend to retain hepatocyte-like features [63,64]. A study comparing patients with these two etiologies highlighted a significantly higher frequency of *TERT* and *CTNNB1* mutations in HCV-related HCC (53.6% and 26.4%, respectively) than in HBV-related (41.7% and 16.7%, respectively) [65]. However, mutations in cancer-related genes are very similar between HCV- and alcohol-related HCC suggesting that the difference observed in the literature between HCV- and HBV-induced HCC is mainly explained by the specific genomic profile of HBV-related HCC [13].

Mutations observed in HCV-related HCC are mainly triggered by an indirect effect of immune-mediated chronic inflammation and by liver damage. Moreover, since HCV has a lifecycle exclusively cytoplasmic, pro-oncogenic events are more likely to be indirect [66]. However, several observations suggest a potential direct role of HCV in the carcinogenic process. Clinical evidence showed that HCC can occur in HCV-infected patients in the absence of cirrhosis, although at a lower frequencies than in cirrhotic patients [67,68]. Furthermore, in vivo studies demonstrated that HCC develop in HCV transgenic mice in the absence of inflammation or fibrosis [69,70]. Overall, these results suggested that the persistence of HCV infection and of the expression of viral proteins could play a role in HCC promotion.

### 3.3. HCV Oncogenic Proteins

The core protein is highly conserved and plays a role in viral assembly and in the modulation of cellular transformation, transcription, apoptosis, lipid metabolism, and oxidative stress (reviewed in [71]) (Figure 2). Mice with liver-specific expression of the core gene developed histological features of chronic hepatitis C infection and HCC, without enhanced liver inflammation [69]. Possible oncogenic mechanisms relies on the properties of the HCV core protein to deregulate the cell cycle and promote cell proliferation by interacting with the proteins p53, p73 and Rb, by activating the Raf1/MAPK and Wnt/β-catenin pathways, by modulating the TGFβ signaling or by increasing telomerase activity [29]. This protein may also induce transformational changes in hepatocytes by upregulating several cellular proteins regulating inflammation such as IL-6, gp130, leptin receptor, and STAT3 [29].

In addition, three non-structural HCV proteins (NS3, NS4B, NS5A) activate different mechanisms that inhibit apoptosis, accelerate proliferation, and induce genomic instability [8]. In vitro experiments highlighted that HCV NS3 can induce cellular transformation directly via its serine protease activity [72,73]. HCV NS4B may play a direct role in cellular transformation in cooperation with the HA-*ras* oncogene [74,75], or may have an indirect contribution to carcinogenesis by inducing endoplasmic reticulum (ER) stress and modulating lipid metabolism [76]. HCV NS5A is part of the HCV RNA replication complex and may promote the establishment of chronic HCV infection and then HCC development via its antiapoptotic activity (Figure 2) [77,78].

However, as all these studies analyzing the potential oncogenic properties of HCV proteins are based on overexpression experiments, they do not reflect the expression levels in HCV-infected cells, and do not take into account the inflammatory environment [8].

## 4. Genomics of Hepatitis D-Related HCC

Around 5% of people infected with HBV worldwide are diagnosed as co-infected with hepatitis D virus (HDV), but this percentage is likely underestimated as HDV diagnosis is not systematically assessed in the majority of developing countries [1]. HDV is a defective virus that requires HBsAg to be pathogenic. Therefore, HDV infection can occur through simultaneous coinfection with HBV or through HDV superinfection of chronic HBV-infected patients [79].

### 4.1. HDV Structure and Natural History of Infection

HDV is a small RNA virus (1.7 kb) with a single-stranded genome contained in a nucleocapsid formed by proteins termed HD antigens (HDAg) [80] and surrounded by a lipid envelope with the three types of HBV surface proteins [81,82]. Once entered in hepatocytes, the HDV genome is released, it translocates to the nucleus, and uses RNA polymerase II to replicate [83]. This replication is based on a Rolling-circle mechanism. A first step generates an anti-genome complementary to the HDV genome and a second step uses this anti-genome to reform the initial genome that will be enveloped in the new virions [83]. An anti-genomic RNA is also the template for the synthesis of two forms of HDAg: the S-HDAg proteins necessary for viral replication and the L-HDAg proteins that allow the assembly of virions by binding to the HBsAg proteins [84].

HDV infection generally results in decreased HBV viral replication, although viral dominance appears to fluctuate over time in patients [85,86]. The mechanism proposed to explain this HDV dominance (reviewed in [79]) are based on HDAg, which induce epigenetic regulation of HBV cccDNA reducing pgRNA transcription, repressing HBV enhancer sequences and decreasing the stability of HBV viral RNAs. Moreover, another hypothesis suggests that HDV virion assembly is favored in hepatocytes where HBV does not replicate but when HBsAg are produced from the integrated forms of the virus [51]. Finally, HDV induces a strong immune response producing interferons that could have an antiviral effect on HBV [87].

HDV infection is considered as the most severe form of chronic viral hepatitis, although it relies on the presence of HBV infection [88]. Progression to cirrhosis is observed in 10–15% of these patients after 2 years and in more than 80% of patients after 30 years [89]. Recent clinical observations suggest that co-infection also increases the risk of HCC compared to HBV infection alone [90,91,92]. As HDV requires the presence of HBV to propagate, it is unlikely that HDV has direct oncogenic properties. However, several events occurring during HDV infection may promote HCC development such as induction of oxidative stress by HDV-related inflammation or interference with DNA methylation by HDV proteins [93].

### 4.2. Genomic Landscape of HDV-Related HCC

Few data are currently available to understand the genomic landscape of HDV-related HCC [92]. The analysis of 5 HDV-HCC has revealed a strong activation of pathways involved in cell-cycle, DNA replication, DNA damage, and DNA repair, and suggested a key role of genetic instability in HDV-related carcinogenesis [94]. Comparison of HDV-HCC with HBV-HCC and HCV-HCC underlined that the molecular mechanisms involved are distinct for each hepatitis virus. A genomic analysis on 76 HCC from Mongolian patients included a high proportion of HDV-HCC (*n* = 28) [95,96] and showed an enrichment of mutations in *SPAT1*, a gene encoding for α-spectrin, in HDV-related HCC [96]. However, *SPAT1* had never been reported as a HCC driver gene in other studies and the impact of HDV in liver carcinogenesis remains to be better elucidated. Interestingly, HBV/HDV related HCC harbored less frequent HBV insertion in cancer drivers gene such as the *TERT* promoter compared to HBV alone related HCC [47].

## 5. Interactions with Exposures to Genotoxic Agents

Exposure to genotoxic agents is responsible of mutational processes that can be identified using new generation sequencing through the analysis of mutational signatures. In particular, Aflatoxin B1 (AFB1) and Aristolochic Acid are associated with two specific mutational signatures observed in HCC [97].

### 5.1. HBV Infection and Aflatoxin B1 Exposure

Aflatoxin B1 (AFB1) is a carcinogenic mycotoxin produced by a fungus called *Aspergillus flavus* that can contaminate cereals and peanuts. Whereas it is almost non-existent in Western countries, 90% of the population of sub-Saharan West Africa is exposed to this toxin [98]. AFB1 is an independent risk factor of HCC development when exposure is high but requires HBV as a cofactor when exposure is low or moderate (reviewed in [99]).

From a molecular point of view, AFB1 is oxidized in cells by CYP450 enzymes to form an unstable metabolite that reacts with DNA and produces adducts on guanine residues. The majority of these adducts are repaired by the nucleotide excision mechanism but the remaining DNA lesions can generate mutations, mostly G-T transversions [100]. The G > T substitution on the *TP53* gene induces the R249S mutation frequently identified in HCC and considered as pathognomonic of AFB1 exposure [98]. Several mechanisms have been suggested to explain how HBV could increase the mutagenic effect of AFB1 (reviewed in [98]): the chronic viral infection may directly increase the activity of CYP450 enzymes and generate oxidative stress, the selection pressure may favor clonal proliferation of cells containing these mutations, and finally the HBx protein deregulating DNA repair by nucleotide excision may impair the repair of DNA adducts. We recently showed that patients with HBV related HCC and AFB1 exposure where younger with more frequent HCC harboring stem cell features compared to HBV related HCC without AFB1 exposure [47]. Furthermore, a genomic characterization of HCC related to AFB1 exposure identified frequent mutations in the *ADGRB1* gene in addition to those described in *TP53*. These HCC contained more neo-antigens and harbored high lymphocyte infiltration and high PD-L1 expression [101].

### 5.2. HBV Infection and Aristolochic Acid Exposure

Aristolochic acid is a highly mutagenic compound contained in plants called *Aristolochia* or *Asarum* [102]. These plants have been used in traditional Chinese medicine for several centuries. The interaction between metabolites of aristolochic acid and DNA generates the formation of adducts on adenine residues and thus leads to the appearance of A-T transversions [103,104]. A recent epidemiological study conducted in Taiwan demonstrated that, since aristolochic acid was prescribed as a treatment for chronic hepatitis B until 2003, 59.4% of patients infected with HBV were exposed to the genotoxic agent [105]. This work also demonstrated a dose-dependent association between aristolochic acid exposure and the development of HCC in HBV-infected patients. This suggests a cooperation between HBV and aristolochic acid promoting carcinogenesis [106].

### 5.3. HDV Infection and Genotoxic Exposures

A recent genomic study on Mongolian HDV-HCC identified an association between HDV infection and mutational signatures suggestive of exposition to alkylating agents, tobacco chewing, 1,8-Dinitropyrene and furan [96].

## 6. AAV as a New Player in Viral-Associated Liver Tumors

The Adeno-associated virus (AAV) is a defective DNA virus responsible of non-pathogenic and frequent infections in the general population (30–80%). However, recurrent clonal integrations of AAV2 have been identified in a subgroup of HCC in 2015, suggesting that this virus could be pathogenic in rare circumstances [107].

### 6.1. AAV Structure and Natural History of Infection in the Liver

AAV is a small, single-stranded DNA virus with an unwrapped capsid [108]. It is a defective virus as it requires the presence of another virus to replicate (an adenovirus or a herpes simplex virus) and otherwise establishes a latent infection [109,110]. Twelve distinct serotypes have been described and AAV2 is the most common in humans [111]. They are considered as non-pathogenic to humans and are widely used in gene therapy as the transfection of recombinant AAV vectors shows a high efficacy and low immunogenicity [112].

Although it has been shown a few years ago that the liver is the main site of infection of AAV viruses [113], little information was available on the natural history of infection until recently. The analysis of non-tumor tissues and liver tumors from a cohort of 1319 patients revealed the presence of AAV DNA in 21% of patients, more frequently in non-tumor tissues than in tumors and mainly in young female patients without cirrhosis. Two main AAV genotypes were identified: AAV2 and a recombined sequence of AAV2/AAV13. Moreover, the study assessed the presence of episomal AAV DNA in 26% of non-tumor tissues, strongly associated with viral RNA expression, suggesting that the virus is transcriptionally active in the liver of these patients. Human Herpes Virus 6 (HHV-6) was suggested to be the main helper virus co-infecting liver tissues during active AAV infection.

### 6.2. AAV-Related Liver Carcinogenesis

Only 8% of the tumors analyzed contained AAV DNA and the proportion was identical between malignant and benign tumors, but the malignant tumors harbored a higher number of copies per cell due to clonal viral integrations. A total of 19 clonal integrations were identified in 2% of the HCC cohort and almost no episomal form of the virus. All HCC were developed on non-cirrhotic liver and without other viral etiologies. Within all published studies, the genes described as altered by somatic AAV insertional mutagenesis were *CCNA2*, *CCNE2*, *TERT*, *TNFSF10*, *MLL4*, and *GLI1/INHBE* [107,114,115,116]. Two mechanisms were described to explain the oncogenic consequences of AAV clonal integrations. First, the integrated AAV sequence frequently contains viral enhancers and transcription factor binding sites [117], leading to the strong overexpression of the oncogene at vicinity (*CCNE1* or *TERT*) [46,107]. Second, viral integrated sequences may induce the synthesis of truncated transcripts due to the use of an alternative Transcription Start Site (TSS) or of the viral poly-A site (for integrations in *CCNA2* and *TNFSF10*, respectively). Although rare, viral integration of AAV in specific regions of the human genome can promote hepatocarcinogenesis in a non-cirrhotic liver, confirming that AAV is associated with HCC occurrence via an insertional mutagenesis mechanism.

Interestingly, a region present in all inserted AAV sequences was recently described as a liver-specific enhancer-promoter element [117]. Although this region is absent in most AAV vectors currently in use, different animal models showed that integrated AAV vectors can induce clonal expansion [118] and play a role in carcinogenesis [119,120]. To date, no liver tumor has been reported in patients treated using AAV vectors, but they should be followed longitudinally to monitor long-term effects and assess the potential risk of HCC development [107,121].

## 7. Conclusions & Perspective

Major advances have been performed on the knowledge of the direct and indirect roles of hepatic viruses during liver carcinogenesis as well as on the description of the specific genomic profiles present in HBV- and HCV-related HCC. However, there is still unmet needs in the field of pathophysiology of virus-related carcinogenesis and in translation in clinical practices. First, few data are available in HDV-related liver carcinogenesis. Moreover, the impact of viral suppression of hepatitis B and viral eradication of hepatitis C on HCC pathogenesis remains to be studied in the future, in order to better understand why malignant transformation still occurs even after control of the etiology. The translation in clinical practice is still limited in terms of prediction of HCC occurrence or early detection of HCC in patients with HBV and HCV infection. New biomarkers based on our knowledge of early events of viral liver carcinogenesis should be developed. Finally, future clinical trials of targeted therapies and immunotherapies need to be adapted on the specificity of each virus-induced carcinogenesis, such as enrichment in specific genetic alterations or expression of viral neoantigens.

## Figures and Tables

**Figure 1 jcm-10-01827-f001:**
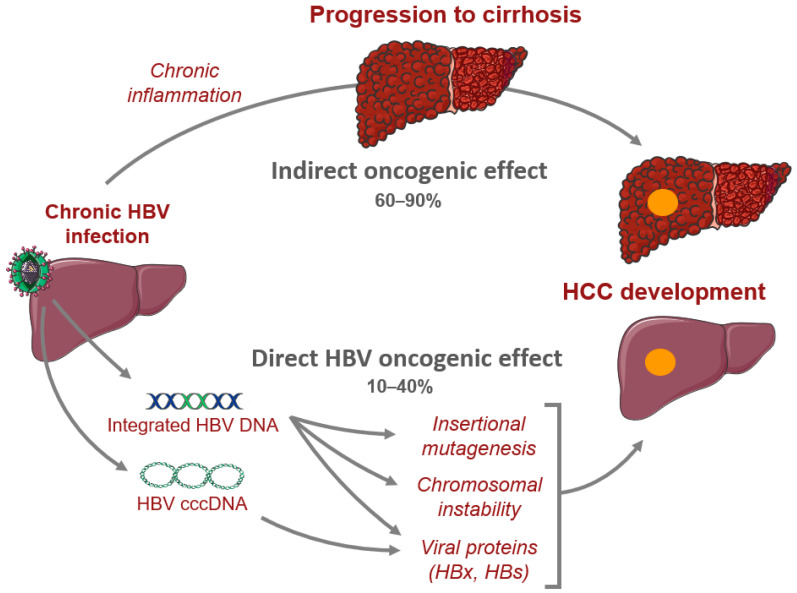
Oncogenic mechanisms related to hepatitis B virus. We figured the direct and indirect mechanisms leading to HCC development in patients with chronic hepatitis B infection.

**Figure 2 jcm-10-01827-f002:**
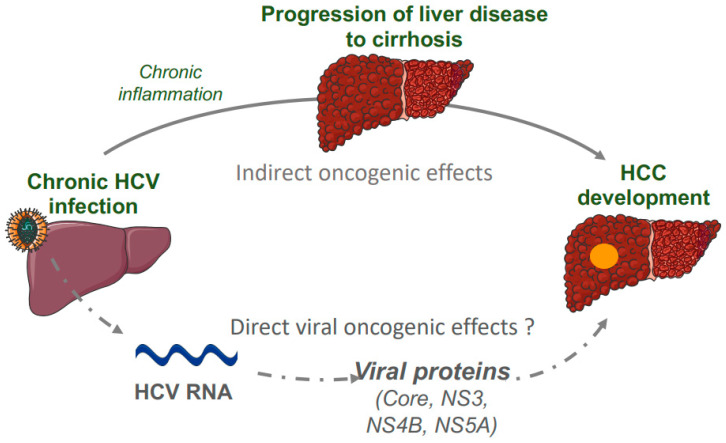
Oncogenic mechanisms related to hepatitis C virus. Direct and indirect oncogenic mechanisms leading to HCC development in patients with chronic hepatitis C infection were represented.

**Table 1 jcm-10-01827-t001:** Description of hepatitis viruses and their relationship with development of hepatocellular carcinoma.

Virus	Genome	Number of Chronic Carriers	Development of Cirrhosis	Development of HCC	Oncogenic Mechanisms	Viral Oncoproteins
HAV	Single-stranded, positive-sense linear RNA virus	No chronic infection	No	No	--	--
HBV	Partially double-stranded, circular DNA virus	257 millions	Yes	Yes	Chronic inflammation, oncoprotein, insertional mutagenesis	HBx, altered HBs or HBx
HCV	Single-stranded, positive-sense linear RNA virus	71 millions	Yes	Yes	Chronic inflammation, oncoprotein	Core protein, NS3, NS4B, NS5A
HDV	Single-stranded, negative sense, closed circular RNA virus	5% of HBV carriers	Yes	Yes	Chronic inflammation, oncoprotein	HDAg
HEV	Single-stranded, positive-sense linear RNA virus	Only in immuno compromized patient	Only in immuno compromized patient	Exceptionnal (immuno compromized patient)	--	--
AAV2	Single-stranded DNA virus	30–80% of positive serology in the general population 26% of AAV2 detected in liver	No	Yes (rare event occurring in normal liver)	Insertional mutagenesis	--

**Table 2 jcm-10-01827-t002:** Genomic profiles of virus-related HCC.

Virus	HCC Features	Enriched Mutated Genes	Main Targeted Genes by Viral Integrations
Hepatitis B	Proliferative HCC, stem cell features, chromosomal instability	*TP53* (R249S), *IRF2*	*TERT*, *MLL4*, *CCNE1*
Hepatitis C	Less agressive, hepatocyte-like features differentiated, chromosomal stability	*TERT promoter*, *CTNNB1*	--
Hepatitis D	Genetic instability	*SPAT1*	--
Adeno-Associated Virus type 2	--	*PTEN*, *AXIN1*	*CCNA2*, *CCNE1*, *GLI1*

## Data Availability

Not applicable.

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
