# Peer review of "Genomics of Viral Hepatitis-Associated Liver Tumors"

_jcm, 2021, doi:10.3390/jcm10091827_

Round 1

Reviewer 1 Report

Reviewing comments.

The manuscript entitled “Genomics of viral hepatitis-associated liver tumors” by Peneau C.et al reviews insightfully the virus-related liver carcinogenesis covering HBV-, HBV+HDV-, HCV- and AAV2-associated liver diseases and liver tumors. These research fields have accumulated enormous knowledge with evidence, but at the same time with a variety of hypotheses and controversy. The manuscript neatly summarizes these outcomes and fairly categorizes the conceivable mechanisms with the critical comments. The mechanistic review on AAV2-associated HCC would be highly informative and contribute to evaluate possible effects of AAV2 integration into host chromosomes in the clinical trials of gene delivery using engineered AAV2. This manuscript could be appreciated by medical researchers and clinicians who are involving in liver diseases and gene therapy. It would be better to be improved in some parts as commented in the followings.

Comments.

  1. Emphasis on chronic inflammation and liver cirrhosis as the major predisposition of HCC would be appreciated.

Not only viral-induced, but also NAFLD- or alcohol-induced chronic inflammation, liver cirrhosis, gives rise to genomic instability, escape from senescence, and dysregulation of some oncogenes. In that sense, persistent viral infection with immunological defense or response associates with chronic inflammation that is the most important common predisposition of HCC. Especially it is the case of the HCV-associated HCC. In the cases of HDV-associated HCC, the incidence of early onset of HCC might be related to severe cirrhosis in younger ages.

  1. On Section 6.

At the introductory part or at the final part, it is critical to point out the putative risk of AAV gene therapy since AAV gene therapy has been regarded as one of the most efficient and safe gene deliveries. There are a few reports of long-time survey in human cases but recently a study was reported with canine cases. They show that AAV integration induced dysregulation of host genes strongly associated oncogenes, but not yet resulted in HCC. The careful evaluation of long-time survey in clinical trials has to be done. It also remains to examine experimentally or prospectively, but not retrospectively, the oncogenic ability of wild AAV2 in animal models.

Reference: Nguyes G. N. et al., (2021) Nature Biotechnology, 39: 47-55.

  1. Minor point related to HBV and HBx

It seems to be better to add some words (bold letters) in the following description.

Line100: This illegimate integration process is …

Line134: the role of HBx in viral replication through degradation of the host factors and transcriptional modulation and … 

HBV integration is illegimate and integration mechanism is not well defined yet.

HBx is a multifunctional regulator and its hijacking function to degrade Smc5/6 is the critical role of HBxseishi78 in HBV infection.

Author Response

Our point-by-point response to the comments of the reviewer:

Reviewers' comments:

Reviewer #1:

The manuscript entitled “Genomics of viral hepatitis-associated liver tumors” by Peneau C. et al reviews insightfully the virus-related liver carcinogenesis covering HBV-, HBV+HDV-, HCV- and AAV2-associated liver diseases and liver tumors. These research fields have accumulated enormous knowledge with evidence, but at the same time with a variety of hypotheses and controversy. The manuscript neatly summarizes these outcomes and fairly categorizes the conceivable mechanisms with the critical comments. The mechanistic review on AAV2-associated HCC would be highly informative and contribute to evaluate possible effects of AAV2 integration into host chromosomes in the clinical trials of gene delivery using engineered AAV2. This manuscript could be appreciated by medical researchers and clinicians who are involving in liver diseases and gene therapy. It would be better to be improved in some parts as commented in the followings.

We warmly thank Reviewer #1 for his/her positive assessment of our work.

Comments.

  1. Emphasis on chronic inflammation and liver cirrhosis as the major predisposition of HCC would be appreciated.

Not only viral-induced, but also NAFLD- or alcohol-induced chronic inflammation, liver cirrhosis, gives rise to genomic instability, escape from senescence, and dysregulation of some oncogenes. In that sense, persistent viral infection with immunological defense or response associates with chronic inflammation that is the most important common predisposition of HCC. Especially it is the case of the HCV-associated HCC. In the cases of HDV-associated HCC, the incidence of early onset of HCC might be related to severe cirrhosis in younger ages.

Thank you for this comment. To emphasize the role of chronic inflammation and liver cirrhosis in HCC initiation and development, we added the following sentence in the introduction (lines 48-54):

“Regardless of the cause (viral infection, chronic alcohol consumption, metabolic syn-drome), chronic inflammation leading to liver cirrhosis plays a crucial role in carcino-genesis as up to 85% of HCC occurs on a background of cirrhosis9. Early oncogenic steps include WNT-β-catenin pathway activation, re-expression of fetal genes, deregulation of protein-folding machinery and the response to oxidative stress, replicative senescence bypassed by TERT promoter mutation9. However, apart from this common indirect mechanism, some studies showed that hepatitis viruses may have direct oncogenic properties, and especially HBV as 10 to 30% of HBV-associated HCC develop in a non-cirrhotic liver10,11.”

  1. On Section 6.

At the introductory part or at the final part, it is critical to point out the putative risk of AAV gene therapy since AAV gene therapy has been regarded as one of the most efficient and safe gene deliveries. There are a few reports of long-time survey in human cases but recently a study was reported with canine cases. They show that AAV integration induced dysregulation of host genes strongly associated oncogenes, but not yet resulted in HCC. The careful evaluation of long-time survey in clinical trials has to be done. It also remains to examine experimentally or prospectively, but not retrospectively, the oncogenic ability of wild AAV2 in animal models.

Reference: Nguyes G. N. et al., (2021) Nature Biotechnology, 39: 47-55.

Thank you for this suggestion. We have added the following paragraph at the end of the 6.2 section (AAV-related liver carcinogenesis – lines 429-435):

“Interestingly, a region present in all inserted AAV sequences was recently described as a liver-specific enhancer-promoter element117. Although this region is absent in most AAV vectors currently in use, different animal models showed that integrated AAV vectors can induce clonal expansion118 and play a role in carcinogenesis119,120. To date, no liver tumor has been reported in patients treated using AAV vectors, but they should be followed longitudinally to monitor long-term effects and assess the potential risk of HCC development107,121.”

  1. Logan, G. J. et al. Identification of liver-specific enhancer–promoter activity in the 3′ untranslated region of the wild-type AAV2 genome. Nat Genet 49, 1267–1273 (2017).
  2. Nguyen, G. N. A long-term study of AAV gene therapy in dogs with hemophilia A identifies clonal expansions of transduced liver cells. Nature Biotechnology 16.
  3. Chandler, R. J. et al. Vector design influences hepatic genotoxicity after adeno-associated virus gene therapy. The Journal of Clinical Investigation 125, 11 (2015).
  4. Donsante, A. et al. AAV Vector Integration Sites in Mouse Hepatocellular Carcinoma. 2.
  5. Nault, J.-C. Wild-type AAV Insertions in Hepatocellular Carcinoma Do Not Inform Debate Over Genotoxicity Risk of Vectorized AAV. 2.

  1. Minor point related to HBV and HBx

It seems to be better to add some words (bold letters) in the following description.

Line100: This illegimate integration process is …

Line134: the role of HBx in viral replication through degradation of the host factors and transcriptional modulation and …

HBV integration is illegimate and integration mechanism is not well defined yet.

HBx is a multifunctional regulator and its hijacking function to degrade Smc5/6 is the critical role of HBxseishi78 in HBV infection.

We added the words in bold letters as suggested to clarify the two sentences (lines 106 and 141-142).

Reviewer 2 Report

This review well summarizes the molecular and genetic factors/pathways in viral hepatitis-associated HCC. 

It covers HBV to HDV and their interactions with genotoxic agents. 

Just one minor suggestion is may be adding a summary table of genomic landscape like table 1 that summarizes each virus's genetic factors.

For example, for HBV, adding a summary of TERT, MLL4, CCNE1 and for HBV, TERT, CTNNB1, etc. 

Author Response

Our point-by-point response to the comments of the reviewer:

Reviewers' comments:

Reviewer #2:

This review well summarizes the molecular and genetic factors/pathways in viral hepatitis-associated HCC.

It covers HBV to HDV and their interactions with genotoxic agents.

We warmly thank Reviewer #2 for his/her positive assessment of our work.

Just one minor suggestion is maybe adding a summary table of genomic landscape like table 1 that summarizes each virus's genetic factors.

For example, for HBV, adding a summary of TERT, MLL4, CCNE1 and for HBV, TERT, CTNNB1, etc.

Thank you for this suggestion. We added a Table 2 to summarize the main altered genes in HCC related to each virus (HBV, HCV, HDV, AAV).